



# Sources and processes of water-soluble and water-insoluble organic aerosol in cold season in Beijing, China

Zhiqiang Zhang[1,2], Yele Sun[1,2,3], Chun Chen[1,2], Bo You[1,2], Aodong Du[1,2], Weiqi Xu[1], Yan Li[1,2], Zhijie Li[1,2], Lu Lei[1,2], Wei Zhou[1], Jiaxing Sun[1,2], Yanmei Qiu[1,2], Lianfang Wei[1], Pingqing Fu[4], Zifa Wang[1,2]

[1]State Key Laboratory of Atmospheric Boundary Layer Physics and Atmospheric Chemistry, Institute of Atmospheric Physics, Chinese Academy of Sciences, Beijing 100029, China

[2]College of Earth and Planetary Sciences, University of Chinese Academy of Sciences, Beijing 100049, China

[3]Collaborative Innovation Center on Forecast and Evaluation of Meteorological Disasters, Nanjing University of Information Science and Technology, Nanjing 210044, China

[4]Institute of Surface-Earth System Science, Tianjin University, Tianjin 300072, China

*Correspondence to*: Yele Sun (sunyele@mail.iap.ac.cn)

**Abstract.** Water-soluble and water-insoluble organic aerosol (WSOA and WIOA) constitute a large fraction of fine particles in winter in northern China, yet our understanding of their sources and processes are still limited. Here we have a comprehensive characterization of WSOA in cold season in Beijing. Particularly, we present the first mass spectral

characterization of WIOA by integrating online and offline organic aerosol measurements from a high-resolution aerosol mass spectrometer. Our results showed that WSOA on average accounted for 59% of the total OA and comprised dominantly secondary OA (SOA, 69%). The WSOA composition showed significant changes during the transition season from autumn to winter. While the photochemical-related SOA dominated WSOA (51%) in early November, the oxidized SOA from biomass burning increased substantially from 8% to 29% during the heating season. Comparatively, local primary OA dominantly from

cooking aerosol contributed the major fraction of WSOA during clean periods. WIOA showed largely different spectral patterns from WSOA which were characterized by prominent hydrocarbon ions series and low oxygen-to-carbon (O/C = 0.19) and organic mass-to-organic carbon ratio (OM/OC = 1.39). The nighttime WIOA showed less oxidized properties (O/C = 0.16 vs. 0.24) with more pronounced polycyclic aromatic hydrocarbons (PAHs) signals than daytime, indicating the impacts of enhanced coal combustion emissions on WIOA. The evolution process of WSOA and WIOA was further demonstrated by the

triangle plot of $f_{44}$ (fraction of *m/z* 44 in OA) vs. $f_{43}$, $f_{44}$ vs. $f_{60}$, and the Van Krevelen diagram (H/C vs. O/C). We also found more oxidized WSOA and an increased contribution of SOA in WSOA compared with previous winter studies in Beijing, indicating that the changes in OA composition due to clean air action have affected the sources and properties of WSOA.

## 1 Introduction

Organic aerosol (OA) is an important component of atmospheric fine particles and exerts significant impacts on air pollution,

public health, radiative forcing and climate change (Jimenez et al., 2009; Tsigaridis et al., 2014). OA can be divided into

primary OA (POA) emitted directly to the atmosphere, e.g., from combustion processes, and secondary OA (SOA) from oxidation of volatile organic compounds (VOCs), or water-soluble OA (WSOA) and water-insoluble OA (WIOA) based on water solubilities of OA. Recent studies showed that SOA formed from both anthropogenic and biogenic VOCs (especially in summer) and POA emitted from biomass burning (especially during winter) are important contributors of WSOA (Sun et al.,

2011a; Bozzetti et al., 2017a; Bozzetti et al., 2017b). WSOA can absorb water due to its hydrophilic properties, affect the optical properties and the ability to act as cloud condensation nuclei, and thus influence the microphysical and radiative properties of clouds (Facchini et al., 1999; Ervens et al., 2005). WSOA can also generate reactive oxygen species and have an adverse effect on human health (Verma et al., 2015). Therefore, WSOA has been widely characterized worldwide in recent years.

Aerodyne Aerosol Mass Spectrometer (AMS) is one of the mostly used instruments for characterization of WSOA (Decarlo et al., 2006; Huang et al., 2010; Sun et al., 2016) because of its capability in measuring high-resolution mass spectra of OA followed by elemental analysis and positive matrix factorization (PMF) analysis. Sun et al. (2011a) performed the first offline-AMS analyses of water-extracted OA in $PM_{2.5}$ and investigated the sources of WSOA in the southeastern US. The results illustrated the different contributions of biogenic SOA and biomass burning emissions to WSOA in different seasons.

Subsequent studies also showed different characteristics of WSOA in different chemical environments. For instance, Xu et al. (2017a) characterized WSOA and the water-solubility by integrating particle-into-liquid-sampler (PILS) and AMS. The results showed that more than 70% of OA was water-soluble at both urban and rural sites in the southeastern US. The PMF analysis revealed much higher water solubilities of SOA factors than POA factors from traffic, cooking and biomass burning. In highly polluted regions, e.g., Kanpur, India, Chakraborty et al. (2017) found that WSOA on average accounted for 55% of OA and

the oxidation level of WSOA (O/C = 0.73) was high. Ge et al. (2017) characterized the WSOA in Yangtze River Delta, China in different seasons. The results showed that OA comprised 60.3% WSOA that was mainly from secondary organic aerosol (68.1%), while the contributions of primary sources to WSOA were small. In 2019, Qiu et al. (2019) investigated the vertical differences in WSOA in Beijing, and found that WSOA showed higher contribution in OA (52%) and higher O/C in city aloft than ground. Also, coal combustion was found to be an important component of WSOA in heating season in Beijing. In addition

to WSOA, several studies also demonstrated the importance of WIOA in OA. For example, Qiu et al. (2019) found that WIOA contributed a larger fraction than WSOA (53% vs. 47%) in winter in Beijing, while Li et al. (2021a) found that WIOA accounted for up to 71% of the total OA in a polluted urban site, Handan. Recent studies found that WIOA played an important role in brown carbon absorption and the contributions were even higher than that of WSOA (Huang et al., 2020; Atwi et al., 2022). However, our understanding of the characteristics of WIOA is still very limited.

In this work, we characterize WSOA in PM$_{2.5}$ samples that were collected in Beijing, China in the cold season from 6

November 2020 to 19 January 2021 mainly using a High-Resolution Time-of-Flight Aerosol Mass Spectrometer (AMS

hereafter). The sources, day-night differences, elemental composition, and the roles of WSOA and WIOA in haze formation

are elucidated. In particular, we present the first spectral characterization of WIOA by integrating the online and offline AMS

measurements, which provides new insights into the sources and composition of WIOA.

## 2 Experimental methods


### 2.1 Sampling site and measurements

Filter sampling was conducted on the roof of a two-floor building at the tower branch of the Institute of Atmospheric Physics

(39°58′28″ N, 116°22′16″ E) in Beijing (Fig. S1a) during the cold season using a high volume sampler (TISCH, USA) at a

flow rate of 1.1 m$^3$ min$^{-1}$. The total of 103 day (08:00–18:00) and night (from 18:30 to 07:30 the next day) samples from 6

November to 5 December 2020, and daily samples (from 08:00 to 07:00 the next day) from 6 December 2020 to 19 January

2021) were collected during this study. After sampling, the filter samples were stored in a freezer (–20 °C) immediately before

the analyses. The details on the measurement site and experimental methods can be found in previous studies (Qiu et al., 2019;

Qiu et al., 2020).

Simultaneously, real-time measurements of non-refractory aerosol species with aerodynamic diameter less than 2.5 μm (NR-

PM$_{2.5}$) including organics (Org), sulfate (SO$_4$), nitrate (NO$_3$), ammonium (NH$_4$), and chloride (Chl) were conducted with a

time-of-flight aerosol chemical speciation monitor equipped with a capture vaporizer (CV-ToF-ACSM, Tofwerk) and the

operations of CV-ToF-ACSM were described in detail in a previous study (Li et al., 2021b). In the middle of this study, i.e.,

from 6 November to 5 December, non-refractory submicron aerosol (NR-PM$_1$) species were also measured by the AMS. The

detailed descriptions of AMS measurements are given in Xu et al. (2022). In addition, the equivalent black carbon (eBC) was

synchronously measured using an Aethalometer (AE-33, Magee Scientific Corp.), and gaseous species including NO$_2$, SO$_2$,

O$_3$ and CO were obtained from the Olympic Center observation site (http://zx.bjmemc.com.cn), which is ~4 km from the

sampling site. The meteorological parameters including temperature ($T$), relative humidity (RH), wind speed (WS) and wind

direction (WD) were obtained from the Beijing meteorological tower at the same site.

### 2.2 Offline analysis

2 – 4 punches (diameter: 24 mm) of filter samples depending on OA mass concentrations were extracted with 40 mL deionized

water (18.3 MΩ). After sonicated for 30 min under low-temperature ice-bath conditions, the extracted solutions were filtered

with 0.45-μm syringe filters (Anpel, PVDF). An aliquot solution was aerosolized with a constant output atomizer using pure

argon (99.999%) that can reduce the influence of $N_2$ on the separation of organic fragments at $m/z$ 28 (mainly $CO^+$, $C_2H_4^+$, and $CH_2N^+$) substantially. The aerosolized particles were dried with a silica gel diffusion dryer followed by a nafion dryer, and then measured by AMS operated in mass-sensitive V-mode. After sample analysis, three deionized water samples were analyzed to reduce the background signals.

Another aliquot was analyzed for water-soluble anions and cations ($SO_4^{2-}$, $NO_3^-$, $NO_2^-$, $Cl^-$, $F^-$, $NH_4^+$, $Na^+$, $K^+$, $Mg^{2+}$ and $Ca^{2+}$) using an ion chromatography (Thermo Scientific, Dionex Aqunion-1100) equipped with IonPac AS11-HC (anion) and IonPac CS12A (cation) chromatographic column systems. The third aliquot was used to measure the WSOC using a total organic carbon (TOC/TN) analyzer (Shimadzu, TOC-L). In addition, organic carbon (OC) and elemental carbon (EC) in filter samples were measured using a Sunset OC/EC analyzer (Sunset Laboratory Inc., Model-4). The detailed descriptions of the analyses of water-soluble ions, WSOC, and OC/EC were given in Qiu et al. (2019).

Figure S2 shows that most of the water-soluble species correlated well with the online-measured species by the CV-ToF-ACSM. The relatively poor correlation of $SO_4^{2-}$ may be caused by a certain amount of organic sulfate (OS) existing in the atmosphere especially during the episodes with high RH (Chen et al., 2019; Wei et al., 2020), because CV-ToF-ACSM cannot separate OS from sulfate. Despite this, most of the online- and offline-measured sulfate is comparable (Fig. S2h). eBC showed an overall higher concentration than EC, which may be caused by the different measuring methods. The reconstructed $PM_{2.5}$ ($PM_{2.5}$_reconstructed = WSOA + WIOA + $SO_4^{2-}$ + $NO_3^-$ + $NH_4^+$ + $Cl^-$ + EC + $Ca^{2+}$ + $F^-$ + $Na^+$ + $NO_2^-$ + $K^+$ + $Mg^{2+}$) showed a tight correlation with $PM_{2.5}$ measured at the Olympic Center monitoring site ($r^2$ = 0.92, slope = 0.96) and the online-measured $PM_{2.5}$ (NR-$PM_{2.5}$ measured by CV-ToF-ACSM plus eBC, $r^2$ = 0.95, slope = 0.95). All these results indicate that the measurements, analyses and quantification of aerosol species in this study are reasonably well.

**2.3 AMS data analysis**

The AMS data were analyzed with Squirrel 1.62F and PIKA 1.22F written in Igor Pro (http://cires1.colorado.edu/jimenez-group/ToFAMSResources/ToFSoftware/index.html). We found that $CO^+$ was highly correlated with $CO_2^+$ ($r^2$ = 0.98, Fig. S3a), and the ratio of $CO^+$ to $CO_2^+$ (0.67) was similar to those (0.65 and 0.72) reported in previous studies (Qiu et al., 2019; Qiu et al., 2020). Therefore, the original $CO^+$ values were used for subsequent elemental analysis and PMF analysis. The organic mass-to-organic carbon (OM/OC), oxygen-to-carbon (O/C), hydrogen-to-carbon (H/C), and nitrogen-to-carbon (N/C) ratios were then determined using the Improved-Ambient (I-A) method (Canagaratna et al., 2015). Note that the O/C ratios were on average 4% lower and the H/C ratios were 3% higher than those determined using the traditional assumptions, i.e., $CO^+$ = $CO_2^+$ (Fig. S4d).

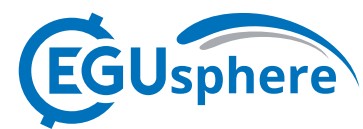

WSOA was simply determined as OM/OC × WSOC. Previously, WIOC (= OC – WSOC) was converted to WIOA using a constant OM/OC ratio of 1.3 (Sun et al., 2011a).

In this study, we had a period of collocated real-time online measurements of ambient OA in $PM_1$ by the same AMS. As a result, WIOA can also be determined as the difference between the ambient OA and WSOA. Because the different size cutoff of online measurements and filter collections, the OA measured by AMS was first scaled to $PM_{2.5}$ OA using a factor of 1.50

according to the inter-comparisons (Fig. S3b). Therefore, WIOA was determined as WIOA = $OA_{PM1}$ × 1.5 – WSOA. Similarly, the high-resolution mass spectra (HRMS) of WIOA, for the first time, were obtained as the difference between the average mass-weighted HRMS of ambient OA by multiplying a factor of 1.5 and those of WSOA measured during the same period of the time. The uncertainty of WIOA was estimated based on the errors of OA (0.43%) and WSOA (0.59%), and the average

uncertainty was 0.73%. However, the uncertainty can be up to 3.4% for the data with low concentrations or the data with low fractions of WIOA.

The I-A elemental analysis showed that the average OM/OC ratio of WIOA was 1.46. The I-A approach was used in field studies mainly due to the underestimation of oxidized organic compounds, however, the high-resolution mass spectra of WIOA showed the dominance of hydrocarbon ion series. We therefore determined the elemental composition of WIOA using the A-

A method and found that the average OM/OC of WIOA was 1.39. WIOA was then estimated as (OC – WSOC) × 1.4, which was 8% higher than that from previous studies. Figure S3c shows that WIOA estimated with a constant OM/OC ratio of 1.4 was highly correlated ($r^2$ = 0.78) with that calculated from the difference between OA and WSOA, and the slope of 0.85 suggested that the quantitation of WIOA is reasonably well. In addition, the reconstructed OA (= WSOA + WIOA) was highly correlated with the total $PM_{2.5}$ OA measured by CV-ToF-ACSM ($r^2$ = 0.94, slope = 0.97, Fig. S2c), further supporting the

reasonable quantification of WSOA and WIOA in this study.

PMF analysis (Paatero and Tapper, 1994) was performed on the high-resolution mass spectra of WSOA. The $m/z$'s up to 150 and specific polycyclic aromatic hydrocarbons (PAHs) fragments (e.g., $C_{12}H_8^+$, $C_{13}H_9^+$, $C_{14}H_{10}^+$, $C_{15}H_9^+$ and $C_{16}H_{10}^+$) were included in the PMF analysis (Ulbrich et al., 2009). The detailed procedures for the data processing and the selection of PMF factors have been given in Zhang et al. (2011). According to the diagnostic plots in Fig. S5, the correlations between OA

factors with tracer species (Figs. S6 and S7), and the mass spectral comparisons with previously resolved WSOA factors, the five-factor solution at fPeak = 0 including three SOA factors, i.e., OOA1, OOA2 and an oxidized BBOA (oBBOA), two POA factors including a coal combustion and biomass burning OA (CBOA) and a local OA (LOA) factor (Fig. 4). Comparatively, PMF analysis of online measurements of OA resolved three POA factors including a fossil fuel OA (FFOA), a BBOA, and a cooking OA (COA), and two SOA factors, i.e., OOA1 and OOA2 (Xu et al., 2022).

We further estimated primary OC (POC) and secondary OC (SOC) using the EC-tracer method (Turpin and Huntzicker, 1995),

in which the lowest ratio of organic carbon to elemental carbon (OC/EC)$_{pri}$ from primary emissions (5.4) was used in this study

(Fig. S8). Then, POC was estimated as POC = EC × (OC/EC)$_{pri}$, and SOC was calculated as the difference between OC and

POC, i.e., SOC = OC – POC. It should be noted that the (OC/EC)$_{pri}$ ratio in this study was relatively higher than previous

works, which might be due to the relatively higher oxidation process during this campaign. POC was converted to POM using

a factor of 1.2 (POM = POC × 1.2, the factor is adopted from the vehicle related POA in Collier et al. (2015)), which has also

been used in previous studies (Sun et al., 2011a; Wang et al., 2016; Qiu et al., 2020), and SOM was estimated as the difference

of OA and POM, i.e., SOM = OA – POM.

**3 Results and discussion**

**3.1 General descriptions of WSOA and WIOA**

The average (± 1σ) PM$_{2.5}$ mass loading during the observation period was 32.6 (± 26.0) μg m$^{-3}$ and the composition was

dominated by OA (39%) and nitrate (27%). The average mass concentration of WSOA and WIOA was 7.5 (± 5.5) μg m$^{-3}$ and

5.1 (± 3.8) μg m$^{-3}$, on average contributing 59% and 41%, respectively to OA. WSOA showed a higher mass fraction in OA

compared with previous studies in Beijing (Qiu et al., 2019). One explanation is the continuous increase of SOA in total OA

during wintertime in recent years (Lei et al., 2021).

WSOA was well correlated with SOM ($r^2$ = 0.92, Fig. S9a), suggesting that WSOA may be dominantly from secondary

formation process in the atmosphere (Zhang et al., 2018). Further support is the good correlations between WSOC and SOC,

and WSOA and SNA (Figs. S9b and S9c). Comparatively, WIOA was highly correlated with POM ($r^2$ = 0.87, Fig. S9d) and

EC ($r^2$ = 0.87, Fig. S9f), and WIOC was highly correlated with POC ($r^2$ = 0.87, Fig. S9e), indicating the major sources of

primary emissions for WIOA and WIOC. The slope of 0.90 for WIOA vs. POM indicated that part of POM was water-soluble,

which was consistent with the results in a previous study (Qiu et al., 2020).

To better understand the variations in WSOA and WIOA, we classified the entire study into four periods (Fig. 1). The first

period (P1) showed high PM$_{2.5}$ concentration (73.1 ± 37.7 μg m$^{-3}$) with a dominance of nitrate (34%) and WSOA (22%). The

second period (P2) occurred at Hanyi festival and showed a large influence of biomass burning (Xu et al., 2022), and the

contributions of WIOA and chloride to PM$_{2.5}$ increased to 18% and 4%, respectively. The third period (P3) was relatively clean

with an average PM$_{2.5}$ concentration of 21.9 (± 9.0) μg m$^{-3}$, and WSOA on average contributed 24% to the total OA. The rest

of period (P4) with an average PM$_{2.5}$ of 28.4 (± 16.6) μg m$^{-3}$, showed comparable contributions of nitrate (25%) and WSOA

(22%). Although the PM$_{2.5}$ composition showed significant changes from P1/P2 to P3/P4 as indicated by large decreases in



nitrate and increases in sulfate, the contribution of WSOA was relatively stable (22–24%) suggesting that WSOA was consistently important during both non-heating and heating period.

Figure 2 shows the fractions of WSOA in OA and elemental compositions of WSOA in this study, and a comparison with the data from previous studies. Previous studies found that OA in summer was more water-soluble than in winter at the same site (Feng et al., 2006; Kondo et al., 2007; Sun et al., 2011a; Du et al., 2014; Cheng et al., 2014; Xiang et al., 2017; Ye et al., 2017; Qiu et al., 2019; Qiu et al., 2020) due to high primary emissions with less oxidation and related PAHs and n-alkanes with low solubility in winter (Capel et al., 1991; Zhang et al., 2008). Indeed, the WSOC/OC in the bin of 0.6–0.7 showed a higher

frequency in summer than winter (Fig. S11b). However, the statistics of multiple datasets in Fig. S11a showed that WSOC had slightly higher fractions in OC in winter (0.40) than summer (0.36), and WSOC/OC ratios fell dominantly within the range of 0.3–0.4 in summer, and 0.4–0.5 in winter (Fig. S11b). These results suggest various factors influencing the water-solubility of OC in different chemical environments. For example, OA can be relatively water-soluble in winter in an environment with large impacts of biomass burning. In this study, the average ratio of WSOA/OA and WSOC/OC was 0.60 and 0.49, respectively,

which was higher than that (WSOA/OA = 0.47) observed at the same site in winter in 2019 (Qiu et al., 2019) and the Northern China Plain in winter in 2015 (WSOA/OA = 0.29) (Li et al., 2021a). The higher water-solubility of OA in this study might be the result of the continuous increase in SOA and the decrease of POA in Beijing in recent years. Consistently, the average OM/OC and O/C were 2.16 and 0.73, respectively, which were also higher than those of WSOA observed previously. We noticed that WSOA in this study also showed relatively high N/C ratio (0.053), indicating the importance of nitrogen-

containing compounds.

**3.2 Sources of WSOA**

Figure 3 shows the high-resolution mass spectra and time series of five WSOA factors from PMF analysis. The first OOA1 showed a consistent variation with temperature and moderately correlated with $C_2H_4O^+$ (*m/z* 44). However, this factor was not correlated with either secondary inorganic species or primary aerosol species. One explanation is that OOA1 was likely from

the partitioning of water-soluble organic compounds, the emissions of which depend on temperature. Similar temperature-related trends of OA have also been observed in previous studies (Leaitch et al., 2011; Daellenbach et al., 2016; Bozzetti et al., 2017b). The second OOA2 is similar to the SV-OOA identified in previous studies (Decarlo et al., 2010; Zhao et al., 2019) and showed a good correlation with $NO_3^-$ that was found dominantly from photochemical processing in winter (Sun et al., 2013b; Chen et al., 2020). The higher concentration of nitrate during daytime than nighttime further supported the dominant formation

of nitrate from photochemical production. Moreover, OOA2 correlated moderately well with hydrocarbon ions and PAH-related ions, suggesting that OOA2 is very likely a factor from photochemical processing of anthropogenic VOCs. Although both OOA1 and OOA2 were characterized by high $CO_2^+$ (*m/z* 44), OOA2 showed slightly higher O/C ratio (0.99 vs. 0.94) and





very different $CO^+/CO_2^+$ ratio (0.56 vs. 0.75). This result indicated that the composition of OOA1 and OOA2 could be very different. LOA is well correlated with the tracer ion of COA ($C_6H_{10}O^+$) (Sun et al., 2011b) and specific nitrogen-containing

ions (e.g., $C_3H_8N^+$ and $C_2H_4N^+$) (Aiken et al., 2009; Sun et al., 2011b; Ye et al., 2017) indicating that this factor is mainly from local emissions. We noticed that the temporal variations of LOA were relatively small throughout the study, consistent with the relatively stable cooking emissions in Beijing. This result further supported that LOA factor could be dominantly from local cooking emissions. The oxidized BBOA (oBBOA) showed good correlations with chloride, and the two marker ions of biomass burning, i.e., $C_2H_4O_2^+$ and $C_3H_5O_2^+$ (Chhabra et al., 2010; Sun et al., 2016). Considering that oBBOA showed a large

increase during the biomass burning event on 15 November 2020 and the small signals of $C_2H_4O_2^+$ in the spectrum, this factor likely represented a SOA factor from oxidation of biomass burning emissions. Compared with previous studies, we were unable to distinguish water-soluble coal combustion and biomass burning OA. Indeed, the CBOA factor in Fig. 3 was well correlated with EC and PAHs (Figs. S6 and S7) and the mass spectrum was characterized by high value of $f_{60}$ indicating that it is a mixed factor from coal combustion and biomass burning. The low O/C ratio (0.31) and high content of $C_xH_y^+$ ions (45%

of total WSOA composition) further supported the primary characteristics of this factor.

SOA (= OOA1 + OOA2 + oBBOA) is the dominant contributor of WSOA (69%), and LOA is the dominant water-soluble POA on average accounting for 23%. Although WSOA and WIOA showed relatively stable contributions throughout the study, the WSOA composition showed substantial changes among different periods. As shown in Fig. 4, the photochemical-related OOA2 constituted the majority of WSOA (51%) during P1, and the oxidized BBOA showed a large increase from 8% to 26%

during the biomass burning period P2. Comparatively, water-soluble POA showed large increases during the heating season, particularly during the relatively clean P3. LOA accounted for more than half of WSOA (56%) during P3. This is consistent with previous studies showing much higher contributions of COA during cleaner periods in Beijing (Sun et al., 2013b). Although the coal combustion and biomass burning-related water-soluble CBOA showed an increase from 6–7% to 10%, we found that the largest increase in WSOA during the heating season was oBBOA. This result suggested that the considerable

fraction of oBBOA in Beijing was likely from regional transport rather than local source emissions.

### 3.3 Mass spectral characterization of WIOA

Figure 5 presents the average mass spectra of WIOA for the entire period, daytime and nighttime, and a comparison with those of WSOA. While the WSOA spectra were characterized by high *m/z* 44 (mainly $CO_2^+$) and O/C (~0.8), and high contents of oxygenated ions of $C_xH_yO_2^+$ and $C_xH_yO_1^+$ (21.3% and 30.9%), the mass spectra of WIOA resembled previously resolved

primary OAs, e.g., HOA and COA, which are characterized by prominent peaks at *m/z* 41, *m/z* 43, *m/z* 55 and *m/z* 57. The average O/C and N/C ratios of WIOA were 0.19 and 0.01, respectively, which are much lower than those of WSOA. Comparatively, the content of $C_xH_y^+$ in WIOA increased significantly up to 69.4%. Therefore, WSOA and WIOA showed



largely different composition. While WSOA comprised more oxygenated organic compounds and N-containing species, WIOA was more contributed by hydrophobic compounds with low oxygen and nitrogen contents (Saxena and Hildemann,

235    1996).

We further compared the mass spectral difference of WIOA between day and night. As shown in Fig. 5, the WIOA spectrum during daytime showed higher $m/z$ 44 and O/C (0.24 vs. 0.16) than that at nighttime, suggesting that WIOA was more oxidized during daytime. Indeed, the spectral difference of night and day showed ubiquitously higher $C_nH_{2n-1}^+$ and $C_nH_{2n+1}^+$ ions at night, while the dominance of $CO^+$, $CHO^+$ and $CO_2^+$ during daytime (Fig. 5i). The concentration difference (Fig. 5j) further

illustrated the more impacts of primary emissions, e.g., traffic and coal combustion at night which was supported by the similar spectrum to previously resolved HOA and pronounced PAHs signals ($C_9H_7^+$, $C_{12}H_8^+$, $C_{13}H_9^+$, $C_{14}H_{10}^+$, $C_{15}H_9^+$ and $C_{16}H_{10}^+$). Compared with WIOA, the WSOA during daytime was slightly more oxidized than nighttime (O/C = 0.79 vs. 0.78). The concentration difference of night-day showed typical characteristics of BBOA with pronounced $m/z$ 60 and 73 (Fig. 5e), indicating that BBOA exerted more impacts on WSOA at nighttime.

The water-soluble and water-insoluble fractions of several specific ions are shown in Figs. 6 and S12. While $C_3H_7^+$ and $C_4H_9^+$ were dominantly contributed by WIOA, the oxygenated ions of $CHO^+$ and $CO_2^+$ showed much higher contributions from WSOA. This result is consistent with our conclusion that WSOA contained more oxygenated compounds while WIOA comprised mainly compounds with carbon and hydrogen. Comparatively, the biomass burning marker ion $C_2H_4O_2^+$ showed higher contributions in WSOA than WIOA suggesting the water-soluble properties of BBOA. We noticed that PAHs showed

comparable contributions in WSOA and WIOA during daytime, and higher contributions in WIOA at nighttime. This result suggested the different sources of PAHs between daytime and nighttime. The PAH-related compounds during daytime had higher water-solubility likely due to the more aging process during the transport, while the locally dominant PAHs at nighttime were characterized by higher water-insoluble fractions.

### 3.4 Evolution of WSOA and WIOA

The triangle plot of $f_{44}$ vs. $f_{43}$ (or $f_{CO2}^+$ vs. $f_{C2H3O}^+$ in Fig. S13a) has been widely used in the previous studies to describe the OA evolution process (Ng et al., 2010). As demonstrated in Fig. 7a, WSOA, online OA and WIOA fell within the different regions in the $f_{44}$ vs. $f_{43}$ plot. WIOA showed relatively lower $f_{44}$ and higher $f_{43}$ than online OA and WSOA, yet was characterized by an evolutionary trend from the left-bottom to the top-left corner. Comparatively, WSOA was mainly located at the top-left corner of the plot with high $f_{44}$. It should be noted that as $f_{44}$ increased from ~0.1 to 0.25, $f_{43}$ of WSOA remained at relatively stable

indicating the different evolutionary mechanism of WSOA from previous ambient OA.

The V-shape plot of $f_{55}$ vs. $f_{57}$ (or $f_{C3H3O^+}$ vs. $f_{C3H5O^+}$ in Fig. S13b) can be used as a diagnostic for the presence of COA and HOA (Mohr et al., 2012). COA was generally characterized by high $f_{55}$ and the ratio of $f_{55}/f_{57}$ (Mohr et al., 2012; Zhao et al., 2019). As shown in Fig. 7b, WSOA, especially water-soluble SOA factors (OOA1 and OOA2) showed relatively small values of both $f_{55}$ and $f_{57}$ and thus fell within the left-bottom region of the plot, indicating that COA was unlikely an important contributor of WSOA. Comparatively, WIOA located at the right-top region showed relatively high $f_{55}$ and $f_{57}$ and the ratio of $f_{55}/f_{57}$ (1.82) was slightly lower than that of LOA (2.37) and those (~2) from previously resolved COA factors. This result indicated that COA could be an important component of WIOA.

$f_{44}$ vs. $f_{60}$ (or $f_{CO2^+}$ vs. $f_{C2H4O2^+}$ in Fig. S13c) was commonly used to investigate the aging process of BBOA in the atmosphere (Cubison et al., 2011). BBOA typically evolves from the right-bottom region with low $f_{44}$ and high $f_{60}$ (e.g., fresh BBOA from straw burning and wood combustion in Fig. 7c) to the left-top region with high $f_{44}$ and low $f_{60}$ (e.g., LV-OOA and SV-OOA with less signature of biomass burning). However, we found that the $f_{60}$ of most WSOA and WIOA was close to that of background value ($f_{60}$ = ~0.3%) in the absence of biomass burning. This result suggested that fresh biomass burning was not an important source of OA in this study. In contrast, the biomass burning event and the factor of water-soluble CBOA showed relatively high $f_{60}$ with different $f_{44}$, indicating that biomass burning was important during specific events or periods.

The evolution processes of WSOA and WIOA are further demonstrated by Van Krevelen diagram (H/C vs. O/C) (Heald et al., 2010). As shown in Fig. 7d, WIOA is located at the left-top corner with overall high H/C and low O/C, and evolves towards the right-bottom region. Comparatively, WSOA located in the middle of Van Krevelen diagram evolves similarly to WIOA, yet with different slopes (–0.42 vs. –0.74). This result indicated the different evolutionary mechanism of WIOA and WSOA. For example, the WIOA is mostly outside the region defined by Ng et al. (2011) and the steeper slope suggested the importance of initial heterogeneous oxidation, and then followed by the addition of carbonyl group evolving along with a shallower slope in the diagram (Ng et al., 2011). Comparatively, the shallow slope of WSOA might indicate that fragmentation of carboxylic acid might be a more important evolution pathway.

### 3.5 Chemically-resolved PM pollution

Figure 8 shows the variations of $PM_{2.5}$ and WSOA composition as a function of $PM_{2.5}$ mass loadings, RH and $O_x$. Nitrate and OOA2 showed the most prominent increases as the increase in $PM_{2.5}$ mass loadings, indicating the importance of photochemical processing during the campaign, while other species showed corresponding decreases. In particular, the contribution of OOA2 to WSOA increased from 23% to 63% as $PM_{2.5}$ increased from 30–60 to 120–150 µg m$^{-3}$, highlighting the significance of OOA2 in WSOA. Comparatively, LOA played the most important role in WSOA during the clean period



(0–30 μg m⁻³). Note that OA and chloride showed considerable increases during the severely polluted period (120–150 μg m⁻³) due to the impact of biomass burning event, which is also supported by a corresponding increase of oBBOA in WSOA.

Similar to previous studies (Sun et al., 2013a; Xu et al., 2017b; Wu et al., 2018; Li et al., 2021b), nearly all species increased as the increase of RH when RH < 80%. One explanation is high RH was generally associated with stagnant meteorological conditions leading to the accumulation of pollutants, and enhanced aqueous-phase reactions. However, we found that the photochemical-related OOA2 showed the most significant increase in WSOA as a function of RH. Figure 1 shows that the RH was overall low throughout the study particularly during the P3/P4 periods (RH < 60% for most of the time), and the high RH (> 80%) occurred dominantly during the severe haze episode on 16–18 November. Therefore, the RH dependence of $PM_{2.5}$ and aerosol composition further highlighted the important role of photochemical reactions in the formation of severe haze pollution. Another support is the changes in $PM_{2.5}$ species and WSOA factors as a function of $O_x$ ($O_3$ + $NO_2$) levels. Although nearly all species showed increases as the increase of $O_x$, the increases in nitrate and OOA2 in $PM_{2.5}$ and WSOA were the most significant, and the nitrogen oxidation ratio (NOR) also increased along with $O_x$ (Fig. S14), further demonstrating the importance of photochemical processing in this study.

**4 Conclusions**

We have a comprehensive characterization of WSOA and WIOA in $PM_{2.5}$ during the transition season from autumn to winter in urban Beijing by combining offline analysis and real-time online measurements. Our results showed that WSOA (59%) accounted for a larger fraction than WIOA (41%) in OA, and comprised mainly secondary OA (69%). The WSOA composition showed significant changes from autumn to winter. While SOA (88%), especially OOA2 (51%) dominated WSOA in early November, the oxidized SOA from biomass burning increased substantially from 8% to 29% during the heating season. Comparatively, POA (62%), especially LOA (56%) dominantly from cooking aerosol accounted for the major portion of WSOA during the clean period. The characteristics of WIOA were analyzed, for the first time, by integrating online and offline AMS measurements. The mass spectra of WIOA showed prominent hydrocarbon ion series with low O/C (0.19) and OM/OC (1.39). Also, the WIOA showed higher contribution of $C_xH_y^+$ (69.4% vs. 32.1%), lower contributions of $C_xH_yO_2^+$ (7.6% vs 21.3%), and $C_xH_yO_1^+$ (19.4% vs. 30.9%) than WSOA. The nighttime WIOA showed less oxidized properties (O/C = 0.16 vs. 0.24) with more pronounced PAHs signals than daytime, indicating the impacts of enhanced coal combustion emissions on WIOA. Further analysis showed increased water-solubility of OA compared with previous studies likely due to the increased oxidation process since the clean air action in 2013. Overall, the results in this study improved our understanding of the sources and processes of water-soluble and water-insoluble organic aerosol during the transition season from autumn to winter in



Beijing, and are also helpful to serve as constraints to investigate the impacts of OA on cloud condensation nuclei and radiative forcing.

**Data availability.** The data in this study are available from the authors upon request (sunyele@mail.iap.ac.cn).

**Author contributions.** YS designed the research. ZZ, CC, BY, AD, WX, and YL conducted the measurements and experiments. ZZ and YS analyzed the data. ZL, LL, WZ, JS, YQ, LW, and PF reviewed and commented on the paper. ZZ and YS wrote the paper.

**Competing interests.** The authors declare that they have no conflict of interest.

**Financial support.** This work was supported by the National Natural Science Foundation of China (41975170, 91744207).

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



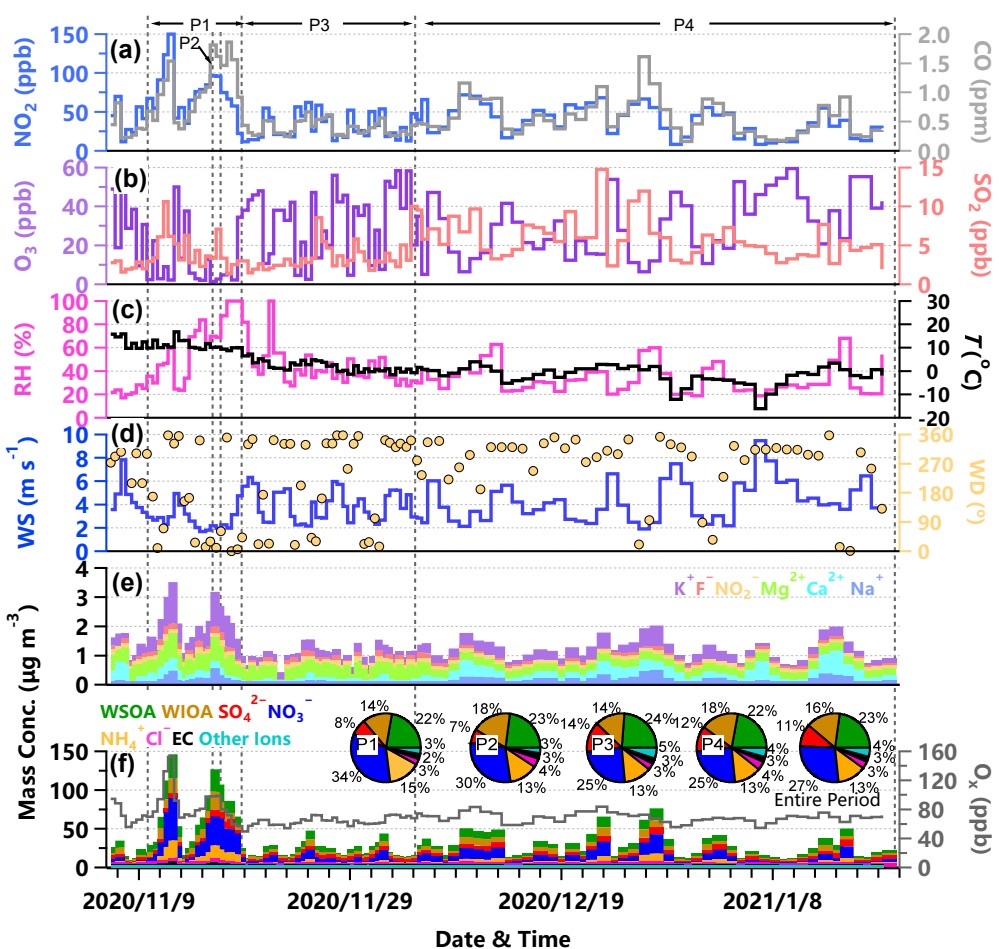

Figure 1. Time series of (a) NO₂ and CO, (b) O₃ and SO₂, (c) relative humidity (RH) and temperature ($T$), (d) wind speed (WS) and wind direction (WD), (e) mass concentration of $K^+$, $F^-$, $NO_2^-$, $Mg^{2+}$, $Ca^{2+}$ and $Na^+$, (f) mass concentration of WSOA, WIOA, $SO_4^{2-}$, $NO_3^-$, $NH_4^+$, $Cl^-$, EC and other ions (sum of $K^+$, $F^-$, $NO_2^-$, $Mg^{2+}$, $Ca^{2+}$ and $Na^+$), also shown in the panel (f) are the offline-measured PM₂.₅ composition pie charts of the four periods and the entire period.




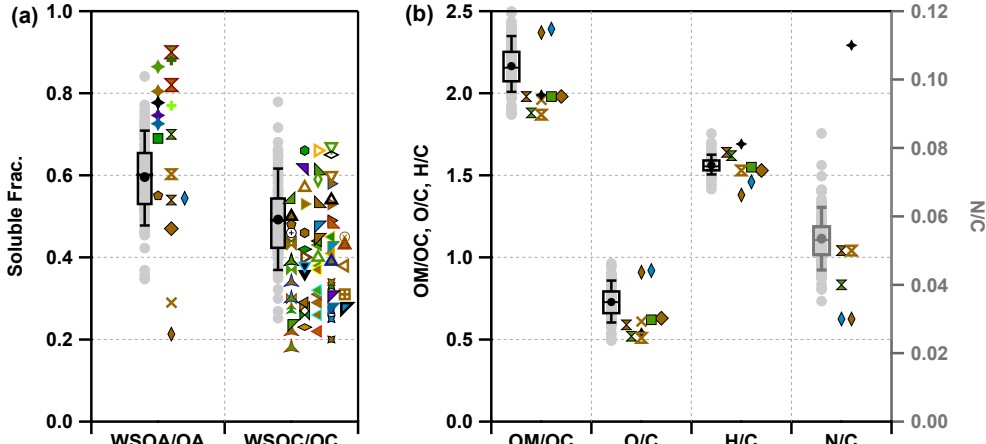

**Figure 2.** Box and whisker plots of (a) the water-soluble fractions of OA (WSOA/OA) and OC (WSOC/OC), (b) OM/OC, O/C, H/C, N/C ratios (I-A method) of WSOA. The circles represent the mean values, the horizontal lines represent the 50th percentiles, the lower and upper of the boxes represent the 25th and 75th percentiles, and the lower and upper whiskers represent the 10th and 90th percentiles in this study. Colored markers represent data values from previous references, data are in purple in spring, green in summer, blue in autumn and brown in winter, the data across the year are in black, the results of different seasons and different sites in the same reference are shown with the different edge colors. The sources of the data were detailed in the Fig. S10.

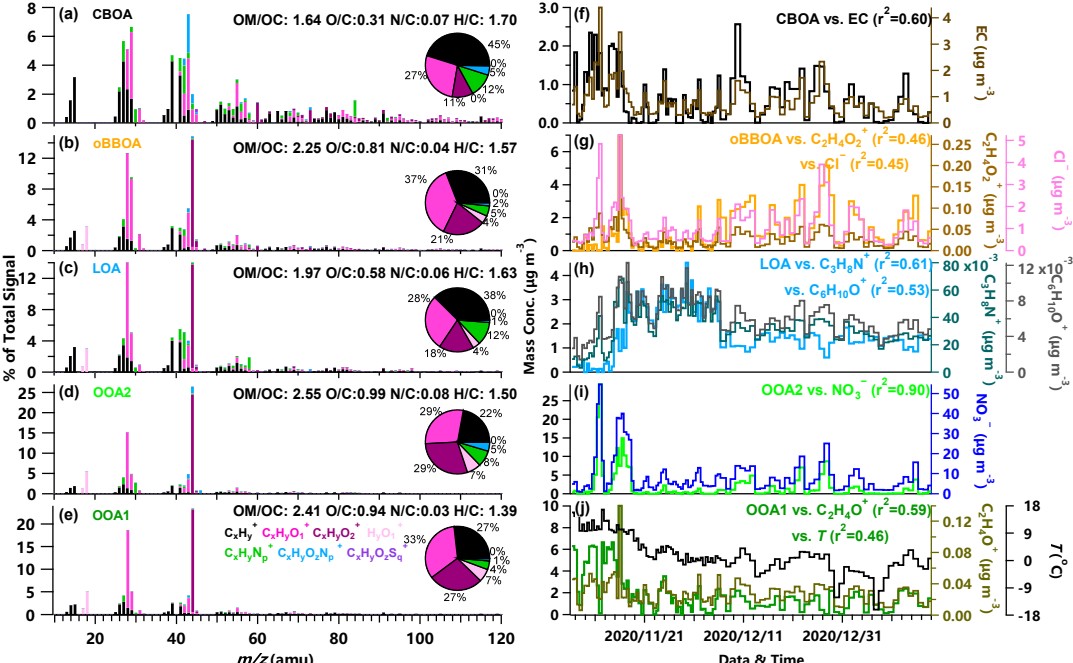

**Figure 3.** (a–e) High-resolution mass spectra of WSOA factors (CBOA, oBBOA, LOA, OOA2 and OOA1) resolved by PMF, (f–j) time series of PMF-resolved WSOA factors, corresponding external tracers and the correlation coefficients between them. Also shown in panel (a–e) are the elemental ratios of WSOA factors that were calculated with the improved-ambient (I-A) method (Canagaratna et al., 2015) and pie charts of the average ions compositions in each factor.



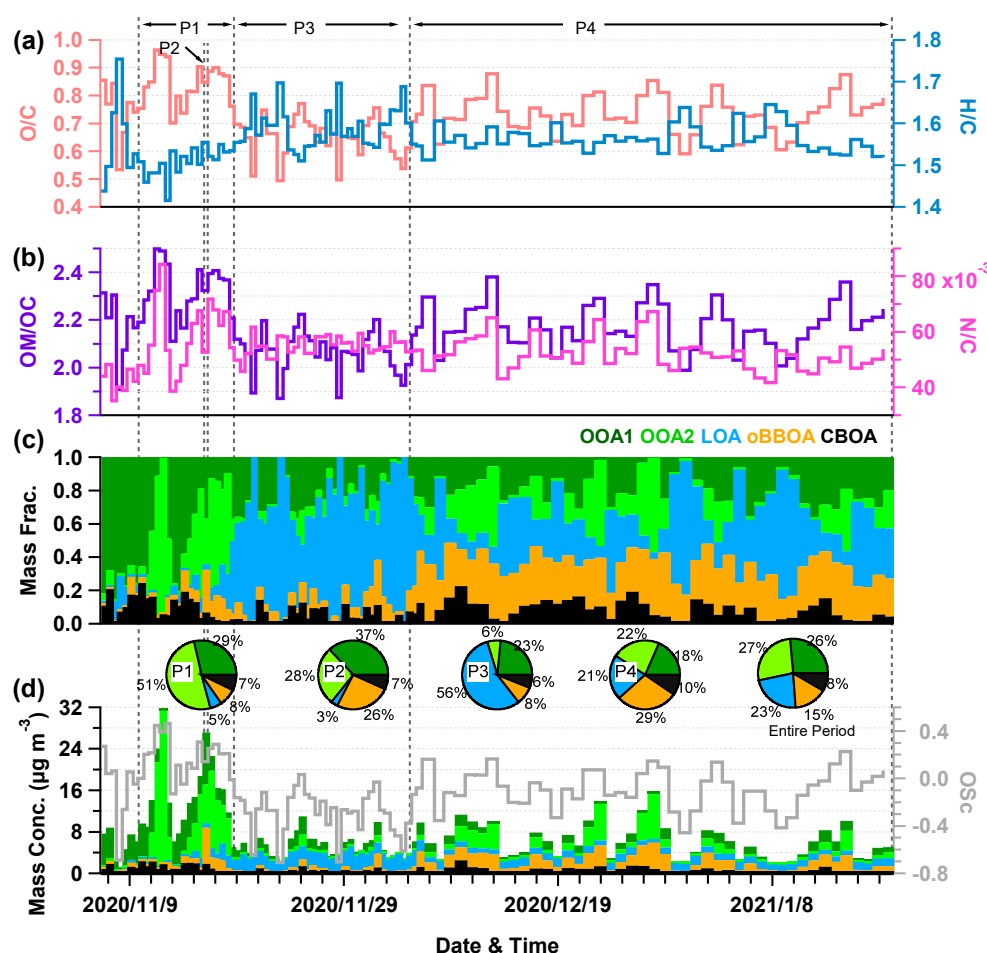

**Figure 4.** Time series of (a, b) elemental ratios of WSOA, (c) mass fractions, (d) mass concentrations of OA factors and oxidation state of WSOA, also shown in panel (d) are the composition pie charts of the four divided periods and the entire period.



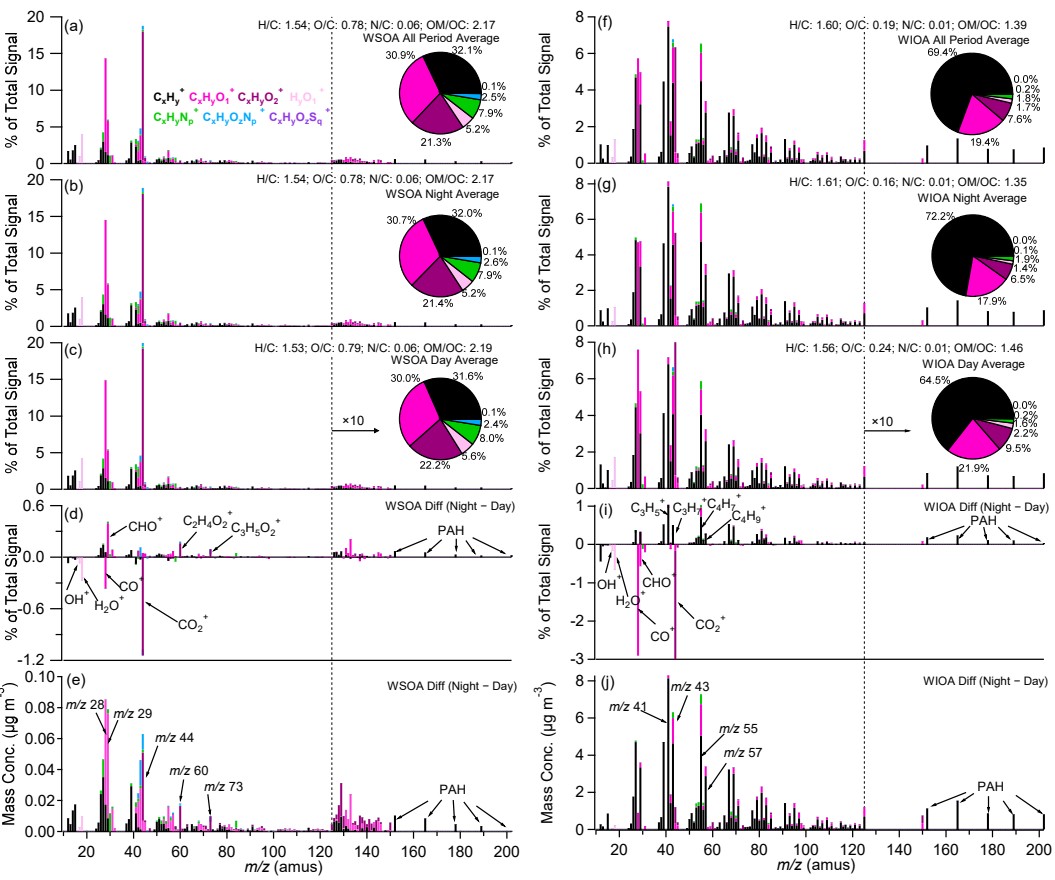

**Figure 5.** Mass spectra of (a, f) all-period, (b, g) night, (c, h) day, and (d, i) night-day differences of signal fraction and (e, j) night-day differences of mass concentration of WSOA and WIOA, respectively, also shown in (a–c) and (f–h) are the ions fraction pie charts of WSOA and WIOA, respectively, note that the value of the mass spectra with *m/z*s larger than 125 are multiplied by 10 for more intuitive display.




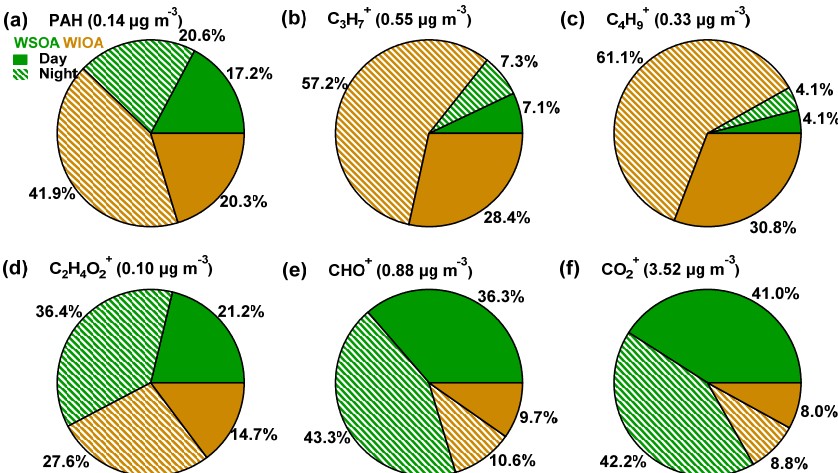

**Figure 6.** Composition of online-measured species (WSOA_day + WSOA_night + WIOA_day + WIOA_night) of (a) PAH, (b) C₃H₇⁺, (c) C₄H₉⁺, (d) C₂H₄O₂⁺, (e) CHO⁺, and (f) CO₂⁺, the shading areas indicate the fraction in the night, while the solid areas indicate the fraction in the day, and the green parts indicate WSOA, while the brown parts indicate WIOA.

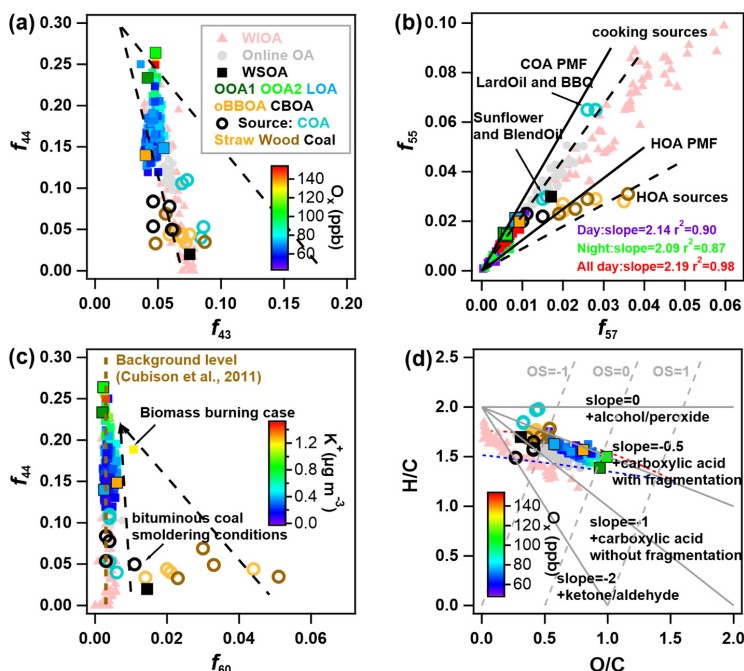

**Figure 7.** Triangle plots of (a) $f_{44}$ vs. $f_{43}$, the dash reference lines are adopted from Ng et al. (2010); (b) $f_{55}$ vs. $f_{57}$, the dashed and solid reference lines are adopted from Mohr et al. (2012); (c) $f_{44}$ vs. $f_{60}$, the dashed background level line and dashed reference lines are adopted from Cubison et al. (2011); and (d) Van Krevelen diagram, the reference lines are adopted from Heald et al. (2010). The data results in the figures contained WSOA (colored squares), online OA (solid grey circles) and WIOA (pink triangles), and the open circles are the source experiment results adopted from Xu et al. (2020). Note that the contributions of water-soluble OOA factors (OOA1 and OOA2) to $f_{55}$ and $f_{57}$ in panel (b) were subtracted.





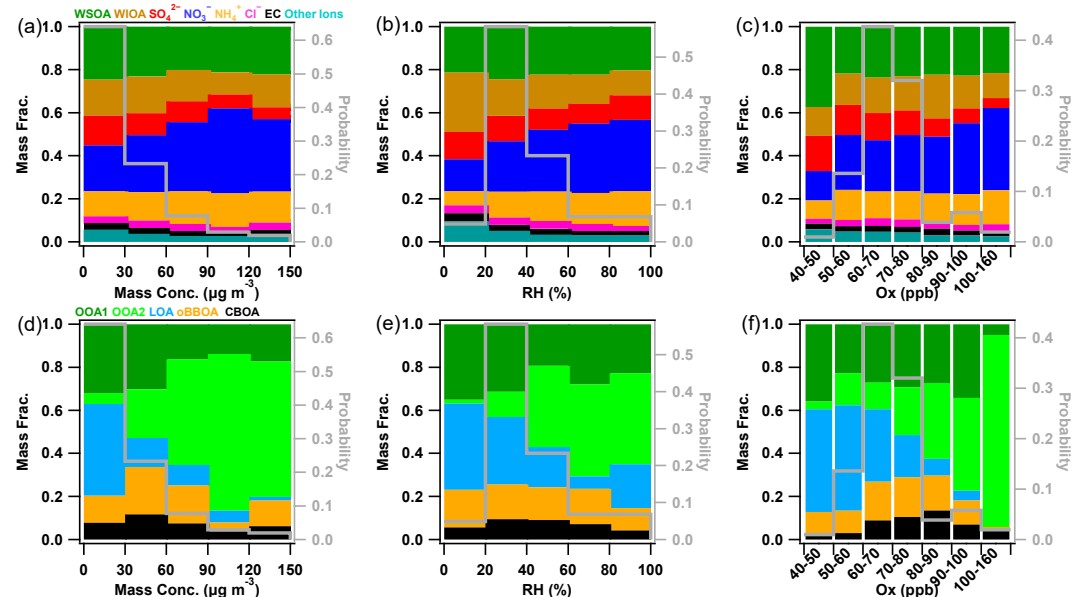

570

**Figure 8. Mass fractions variations of aerosol species in PM$_{2.5}$ as well as OA factors in WSOA as a function of (a, d) mass concentration, (b, e) RH and (c, f) O$_x$.**