# Peer review of "Sources and processes of water-soluble and water-insoluble organic aerosol in cold season in Beijing, China"

_EGUsphere, 2022_

## Author Comment (AC1)

**Response to the Reviewers' comments**

We are thankful to the two referees for their thoughtful and constructive comments which help improve the manuscript substantially. Following the reviewers' suggestions, we have revised the manuscript accordingly. Listed below are our point-by-point responses in blue to each comment that is repeated in italic.

**Response to Reviewer #1**

*General Comments:*

*The manuscript by Zhang et al. characterized water-soluble organic aerosol (WSOA) and water-insoluble organic aerosol (WIOA) using a high-resolution aerosol mass spectrometer (AMS) in the cold season in Beijing. The sources, day-night differences, elemental composition, and the roles of WSOA in haze formation were investigated. They found that WSOA accounted for a larger fraction than WIOA (59% vs. 41%) in OA, and comprised the major part of SOA (69%). Particularly, they presented, for the first time, a study of the high-resolution mass spectra of WIOA by integrating simultaneous online and offline AMS measurements. The WIOA was characterized by prominent hydrocarbon ions series and low oxygen-to-carbon (O/C), which provides new insights into the sources and composition of WIOA in urban Beijing. Overall, this manuscript is well written and I recommend it for publication after minor revisions.*

We thank the reviewer's comments and have revised the manuscript accordingly.

*Specific Comments:*

*1. Because WIOA was obtained from the difference between OA and WSOA. The uncertainty of WIOA depends on the fraction of WIOA in OA, and could be very high. However, the uncertainty of 3.4% in line 125 seems not reasonable, could the authors show more details on the changes of uncertainties of WIOA as a function of the concentrations or mass fractions of WIOA?*

We thank the reviewer's comments. We recalculated the uncertainty of WIOA with the suggestions of the two reviewers by considering the uncertainty of OA, WSOA, the size cutoff correction from Org in $PM_1$ to Org in $PM_{2.5}$ and the uncertainty of the artifacts from the sampling, water extraction and other experiment operations, and added uncertainties variation of WIOA as a function of the mass fractions of WIOA in Fig. R1.

We first fit Error_OA/OA (the ratio of OA error mass concentration to OA mass concentration) as a function of OA mass concentration as shown in Fig. R1a, and the result is: Error_OA/OA = $0.10+3.42×OA^{-0.91}$, then we apply the relationship to OA and WSOA mass concentration to calculate Error_OA/OA, Error_OA, Error_WSOA/WSOA (the ratio of WSOA error mass concentration to the WSOA mass concentration) and Error_WSOA. We then calculate Error_Err_Org_$PM_1$/Err_Org_$PM_1$ and Err_Org_$PM_1$, i.e., the uncertainty brought by the size cutoff correction from Org in $PM_1$ to Org in $PM_{2.5}$ as shown in Fig. S3b and corresponding error mass concentration, here we assume Error_artifacts, i.e., the error of artifacts during the sampling, water extraction and other experiment operation process, to be approximately 10% of the WIOA concentration, and we finally calculate the

uncertainty of WIOA by:

$$Error\_WIOA = \sqrt{Error\_OA^2 + Error\_WSOA^2 + Error\_Err\_Org\_PM_1{}^2 + Error\_artifacts^2}$$, with

this method, the average uncertainty of WIOA is 10.11%, and we revised the corresponding content
as follows:

"The uncertainty of WIOA was estimated based on the error propagation of the uncertainties of OA,
WSOA, the conversion of $PM_1$ to $PM_{2.5}$, and the artifacts during the sampling, water extraction and
other experiment operations (approximately 10%). Our analysis showed that the average uncertainty
of WIOA was 10.1%, and the uncertainty can be up to 12.1% for the data with low concentrations or
the data with low fractions of WIOA."

[Figure]

Figure R1. The uncertainties of (a) OA and (b) WSOA from AMS data analysis versus OA and WSOA
mass concentrations, respectively. Panel (c) shows the uncertainty from the size cutoff correction
from Org in $PM_1$ to Org in $PM_{2.5}$ as a function of the error mass concentration brought by the varying
fitting slope, in which Err_Org_$PM_1$ indicates the error mass concentration of the fitting process.
Panel (d) and (e) show the uncertainties of WIOA as a function of mass concentrations and mass
fraction of WIOA, respectively, the uncertainties are calculated from those of OA, WSOA, the size
cutoff correction, and the artifacts, respectively.

2.  *Did the authors collect blank filters and analyze them with the AMS?*

We have collected and analyzed two blank filters in this experiment, and we have removed the
influence of the background values in the blank filters in the analysis of WSOC, OC/EC and IC, and we
added this description in the revised manuscript as follows:

"The total of 103 day (08:00–18:00) and night (from 18:30 to 07:30 the next day) samples from 6
November to 5 December 2020, and daily samples (from 08:00 to 07:00 the next day) from 6
December 2020 to 19 January 2021) with two additional blank filters were collected during this
study."

The WSOA in this study was estimated indirectly from the measurements of WSOC and OM/OC of
WSOA, i.e., WSOC × OM/OC (Sun et al., 2011; Qiu et al., 2019; Qiu et al., 2020). We did not consider
the influences of blank filters in the AMS analyses, because the average mass concentrations of blank
filters are much lower than those of most samples, and the influences on the determination of
OM/OC ratio are negligible.

3.  How were *PAHs* determined? Please give more details in the measurement or data analysis
section.

We determined PAHs in the same way as that in Dzepina et al. (2007) and Sun et al. (2016), in which PAHs refer as the sum of PAH-related ions (i.e. $m/z$ 115, 128, 152, 165, 178, 189, 202, 215, 226, 239, 252, 276, 300, 326, 350) of the UMR data. In the high-resolution mass spectra analysis, we used the PAH-related fragment ions, i.e., $C_9H_7^+$, $C_{12}H_8^+$, $C_{13}H_9^+$, $C_{14}H_{10}^+$, $C_{15}H_9^+$ and $C_{16}H_{10}^+$ as surrogates. We have added the description and citation in the revised manuscript as follows:

"We determined PAHs in the same way as that in Dzepina et al. (2007) and Sun et al. (2016), in which PAHs refer as the sum of PAH-related ions (i.e., $m/z$ 115, 128, 152, 165, 178, 189, 202, 215, 226, 239, 252, 276, 300, 326, 350) of the UMR data. In the high-resolution mass spectra analysis, we used the PAH-related fragment ions, i.e., $C_9H_7^+$, $C_{12}H_8^+$, $C_{13}H_9^+$, $C_{14}H_{10}^+$, $C_{15}H_9^+$ and $C_{16}H_{10}^+$ as surrogates."

4. The N/C (line 189) and O/C (line 243) ratios were different from those presented in Fig.5. Please have a check.

The N/C ratio of 0.053 in line 189 are the mean values of the N/C ratio time series as summarized in Table S1, and the value is close to the value of 0.06 presented in Fig. 5, when calculating the elemental ratios in Fig. 5, we first averaged the $Number_{samples} \times Number_{m/z}$ Org matrix into a $1 \times Number_{m/z}$ wave, and then calculated the elemental ratios. The values calculated from the two methods are very close or identical within uncertainty as stated by reviewer #2, therefore, we have deleted the corresponding content about the O/C ratio day-night difference.

5. The high $f_{55}/f_{57}$ (2.37) of LOA in line 266 could also support that LOA factor was likely from local cooking emissions. I suggest that a part of this discussion should be presented in the section 3.2.

We thank the reviewer's suggestion. We have added the corresponding discussion in section 3.2 in the revised version of manuscript as follows:

"The high $f_{55}/f_{57}$ (2.37) of LOA as shown in the triangle plot of $f_{55}$ versus $f_{57}$ in Fig. 7b could also support that LOA factor was likely from local cooking emissions"

**Response to Reviewer #2**

General Comments:

Zhang et al. present measurements of WSOA and the WIOA in Beijing during the cold season. They do this using a combination of online and filter-based measurements. Their measuring suite is very complete and their analysis thorough. I would suggest accepting the manuscript as is but have a few small comments:

We thank the reviewer's comments and have revised the manuscript accordingly.

Specific Comments:

1. Line 88: The authors use argon to atomize extracted filter samples. I understand that this reduces interferences by $N_2$ and $O_2$ fragments during AMS measurements, however, it is not a common

*practice in AMS or ACSM use. The author should cite a relevant reference on the use of Argon and effects it may have on quantification. The authors could also expand on this experimental method, for example, if any of the instrument's calibrations need to be adjusted for the use of a different carrier gas.*

We thank the reviewer's suggestion, and we have cited a reference in the revised manuscript (Bozzetti et al., 2017). We used the offline-AMS analysis to obtain the elemental ratios of the samples other than mass concentrations directly, we then obtained the WSOA mass concentration by OM/OC (measured with AMS) multiplying WSOC (measured with TOC analyzer), so the calibration process has no effect on the elemental ratio results. Argon can be used to reduce the influence of $N_2$ on $m/z$ 28 and can better separate $CO^+$ from other fragments at $m/z$ 28. We also calculated the elemental ratios assuming $CO^+ = CO_2^+$ as those in previous ambient studies, which showed small differences as those determined in this study. We have added the effect of Argon in the revised manuscript as follows:

"An aliquot solution was aerosolized with a constant output atomizer using pure argon (99.999%) that can reduce the influence of $N_2$ on the separation of organic fragments at $m/z$ 28 (mainly $CO^+$, $C_2H_4^+$, and $CH_2N^+$) substantially (Bozzetti et al., 2017). Note that the O/C ratios were on average 4% lower and the H/C ratios were 3% higher than those determined using the traditional assumptions, i.e., $CO^+$ = $CO_2^+$ (Fig. S4d)."

2.  *Line 120: The authors calculate WIOA by multiplying the AMS OA(PM$_1$) by 1.5 and substracting the WSOA obtained from offline PM$_{2.5}$ filter collections. They argue that 1.5 is a good number based on the slope of the offline and online measurements presented in figure S3b. I would argue that this slope is driven by some rather high concentration points and that the uncertainty in this slope is rather large. The authors should calculate the uncertainty in this slope and apply it in their error calculations.*

We thank the suggestion of the reviewer, and we have considered the uncertainty of the fitting and size cutoff correction process as shown in Fig. R1c. We then calculated Error_Err_Org_PM$_1$/Err_Org_PM$_1$ and Err_Org_PM$_1$, i.e., the uncertainty brought by the size cutoff correction from Org in PM$_1$ to Org in PM$_{2.5}$ (Fig. S3b) and corresponding error mass concentration with the relationship: Error_OA/OA = 0.10+3.42×OA$^{-0.91}$ as shown in Fig. R1a. The uncertainty from the size cutoff correction process from Org in PM$_1$ to Org in PM$_{2.5}$ is 5.87% and the corresponding error mass concentration is 0.03 µg m$^{-3}$.

3.  *Line 124-125: The errors in WIOA are likely much larger than represented in these sentences. There will be significant error in the slope used to account for PM$_1$-PM$_{2.5}$ differences. The authors should explain how this error is calculated and incorporate the uncertainty in the slope of figure S2b into their error calculations.*

We thank the suggestion of the reviewer, and we have considered the uncertainty of the fitting and size cutoff correction process and recalculated the uncertainty of WIOA by considering the uncertainty of OA, WSOA and the size cutoff correction from Org in PM$_1$ to Org in PM$_{2.5}$ as shown in Fig. R1.

We first fit Error_OA/OA (the ratio of OA error mass concentration to OA mass concentration) as a function of OA mass concentration as shown in Fig. R1a, and the result is: Error_OA/OA =

0.10+3.42×OA$^{-0.91}$, then we apply the relation to OA and WSOA mass concentration to calculate Error_OA/OA, Error_OA, Error_WSOA/WSOA (the ratio of WSOA error mass concentration to the WSOA mass concentration) and Error_WSOA. We then calculate Error_Err_Org_PM$_1$/Err_Org_PM$_1$ and Err_Org_PM$_1$, i.e., the uncertainty brought by the size cutoff correction from Org in PM$_1$ to Org in PM$_{2.5}$ as shown in Fig. S3b and corresponding error mass concentration, here we assume Error_artifacts, i.e., the error of artifacts brought by the sampling, the water extraction and other experiment operation process to be approximately 10% of the WIOA concentration, and we finally calculate the uncertainty of WIOA by:

$$Error\_WIOA = \sqrt{Error\_OA^2 + Error\_WSOA^2 + Error\_Err\_Org\_PM_1{}^2 + Error\_artifacts^2}$$, with

this method, the average uncertainty of WIOA is 10.1%, and we revised the corresponding content as follows:

"The uncertainty of WIOA was estimated based on the error propagation of the uncertainties of OA, WSOA, the conversion of PM$_1$ to PM$_{2.5}$, and the artifacts during the sampling, water extraction and other experiment operations (approximately 10%). Our analysis showed that the average uncertainty of WIOA was 10.1%, and the uncertainty can be up to 12.1% for the data with low concentrations or the data with low fractions of WIOA."

4.  Line 238: The bigger presence of alkyl fragments at night could also be due to repartitioning due to temperature effects, not just differences in oxidation.

We thank the suggestion of the reviewer and added the corresponding context in the revised manuscript as follows:

"in addition, the bigger presence of alkyl fragments at night could also be due to repartitioning for lower temperature at night."

5.  Line 242: I would say these ratios are identical within uncertainty.

We agree with the reviewer and deleted this sentence in the revised manuscript.

**References**

Bozzetti, C., Sosedova, Y., Xiao, M., Daellenbach, K. R., Ulevicius, V., Dudoitis, V., Mordas, G., Byčenkienė, S., Plauškaitė, K., Vlachou, A., Golly, B., Chazeau, B., Besombes, J. L., Baltensperger, U., Jaffrezo, J. L., Slowik, J. G., El Haddad, I., and Prévôt, A. S. H.: Argon offline-AMS source apportionment of organic aerosol over yearly cycles for an urban, rural, and marine site in northern Europe, Atmos. Chem. Phys., 17, 117-141, https://doi.org/10.5194/acp-17-117-2017, 2017.

Dzepina, K., Arey, J., Marr, L. C., Worsnop, D. R., Salcedo, D., Zhang, Q., Onasch, T. B., Molina, L. T., Molina, M. J., and Jimenez, J. L.: Detection of particle-phase polycyclic aromatic hydrocarbons in Mexico City using an aerosol mass spectrometer, International Journal of Mass Spectrometry, 263, 152-170, https://doi.org/10.1016/j.ijms.2007.01.010, 2007.

Qiu, Y., Xu, W., Jia, L., He, Y., Fu, P., Zhang, Q., Xie, Q., Hou, S., Xie, C., Xu, Y., Wang, Z., Worsnop, D. R., and Sun, Y.: Molecular composition and sources of water-soluble organic aerosol in summer in

Beijing, Chemosphere, 255, 126850, https://doi.org/10.1016/j.chemosphere.2020.126850, 2020.

Qiu, Y., Xie, Q., Wang, J., Xu, W., Li, L., Wang, Q., Zhao, J., Chen, Y., Chen, Y., Wu, Y., Du, W., Zhou, W., Lee, J., Zhao, C., Ge, X., Fu, P., Wang, Z., Worsnop, D. R., and Sun, Y.: Vertical characterization and source apportionment of water-soluble organic aerosol with high-resolution aerosol mass spectrometry in Beijing, China, ACS Earth Space Chem., 3, 273-284, https://doi.org/10.1021/acsearthspacechem.8b00155, 2019.

Sun, Y., Zhang, Q., Zheng, M., Ding, X., Edgerton, E. S., and Wang, X.: Characterization and source apportionment of water-soluble organic matter in atmospheric fine particles (PM$_{2.5}$) with high-resolution aerosol mass spectrometry and GC–MS, Environ. Sci. Technol., 45, 4854-4861, https://doi.org/10.1021/es200162h, 2011.

Sun, Y., Du, W., Fu, P., Wang, Q., Li, J., Ge, X., Zhang, Q., Zhu, C., Ren, L., Xu, W., Zhao, J., Han, T., Worsnop, D. R., and Wang, Z.: Primary and secondary aerosols in Beijing in winter: sources, variations and processes, Atmos. Chem. Phys., 16, 8309-8329, https://doi.org/10.5194/acp-16-8309-2016, 2016.